# Supramolecular Fractal Growth of Self-Assembled Fibrillar Networks

**DOI:** 10.3390/gels7020046

**Published:** 2021-04-14

**Authors:** Pedram Nasr, Hannah Leung, France-Isabelle Auzanneau, Michael A. Rogers

**Affiliations:** 1Department of Food Science, University of Guelph, Guelph, ON N1G 2W1, Canada; pnasr@uoguelph.ca (P.N.); hleung07@uoguelph.ca (H.L.); 2Department of Chemistry, University of Guelph, Guelph, ON N1G 2W1, Canada; fauzanne@uoguelph.ca

**Keywords:** 1,3:2,4-Dibenzylidene sorbitol, self-assembled fibrillar networks (SAFiNs), fractality, Cayley Tree, fractal dimension, solvent viscosity, supercooling, crystallization

## Abstract

Complex morphologies, as is the case in self-assembled fibrillar networks (SAFiNs) of 1,3:2,4-Dibenzylidene sorbitol (DBS), are often characterized by their Fractal dimension and not Euclidean. Self-similarity presents for DBS-polyethylene glycol (PEG) SAFiNs in the Cayley Tree branching pattern, similar box-counting fractal dimensions across length scales, and fractals derived from the Avrami model. Irrespective of the crystallization temperature, fractal values corresponded to limited diffusion aggregation and not ballistic particle–cluster aggregation. Additionally, the fractal dimension of the SAFiN was affected more by changes in solvent viscosity (e.g., PEG200 compared to PEG600) than crystallization temperature. Most surprising was the evidence of Cayley branching not only for the radial fibers within the spherulitic but also on the fiber surfaces.

## 1. Introduction

Fractal or self-similar objects exhibit ‘never-ending’ identical patterns across different length scales leading to equal Hausdorff dimensions, often termed fractal dimensions (D). The Hausdorff dimensions of uniform objects—a point = 0, line = 1, square = 2, and cube = 3—are defined as topological dimensions. More complex shapes, such as the Koch Snowflake [1] (D_f_ = 1.26 (D = log 4/log 3) or Sierpinski Carpet [2] (D_f_ = 1.89 (D = log 8/log 3)) (Figure 1), are better defined by their properties of self-similarity and non-Euclidean dimensions. Uniform objects have D_f_ = d, while partly-filled, more open structures where density decreases radially have D_f_ < d [3]. Self-similarity is achieved when each part of a geometric figure has the same statistical character as the whole. Fractality is reported for numerous materials—including, but not limited to, frost [4], fat crystal networks [5], self-assembled polymers [6], and molecular gels [7,8,9,10] are routinely found in nature [11].

Understanding and ultimately controlling the fractal nature of self-assembled fibrillar networks (SAFiNs) is particularly important because the crystalline network morphology determines the macroscale properties (e.g., oil binding, elasticity, and breaking properties) and, ultimately, gel applications [12,13,14]. Molecular self-assembly constructs precision materials, where their supramolecular structures assemble molecule-by-molecule, by way of “bottom–up” nanofabrication, and remarkably, coding for assembly is embedded in the structural motifs of the molecule [15]. Structural motifs direct self-assembly via non-covalent interactions (i.e., hydrogen bonding [16], π–π stacking [17], and van der Waals interactions [18]). Understanding molecular coding (i.e., molecular chirality [19,20,21,22,23,24,25], positional isomers [26,27,28,29], molecular polarity [30,31,32]) is an active area of inquiry for low-molecular-mass (MW < 2000 Da) organogelators (LMOGs) [21,33,34,35,36,37,38,39]. These highly specific interactions drive inter-molecular interactions promoting 1-dimensional (1D) growth—the precursor to gel formation.

However, LMOGs self-assemble in highly dilute environments and factors governing solubility directly contrast forces that control epitaxial growth into axially symmetric elongated aggregates [40,41,42,43,44]. Thus, aspects of solvent are equally important as the molecular coding of the LMOG, making studies that examine the meticulous balance between the gelator and solvent instrumental in developing our fundamental understanding of self-assembly. Herein, the influence of solvent viscosity on SAFiN fractal growth is characterized.

## 2. Results and Discussion

The nano- and micro-structure of SAFiNs, composed of 2,4-Dibenzylidene sorbitol (DBS) in polyethylene glycol (PEG) or poly (propylene glycol) (PPG), depends on both supersaturation and PEG polymer length [45,46,47,48]. LMOGs aggregate via stochastic nucleation events, upon cooling, due to supersaturation [49,50]. When subsequent crystal growth is limited to 1D, the result is extraordinarily high aspect ratio fibers and SAFiNs. DBS/PPG organogels are reported to form spherulite-like morphologies [46], which appear as Maltese-crosses when observed under low magnification cross-polarized light (Figure 2). The precursor to nucleation in a dilute solution is diffusion and phase separation of LMOGs, and a greater solution viscosity increases the kinetic barrier, slowing diffusion and, in practice, leads to greater supersaturation before nucleation, resulting in more nuclei. Consequently, more nuclei translate to larger crystal surface areas for subsequent growth and the spherultie radius decreases. The lower viscosity PEG200 (Figure 2 Top) has fewer, larger (656 ± 129 µm) spherulites compared to PEG600 (302 ± 99 µm), irrespective of concentration, while the spherulite radius is greater for 5 wt.% DBS compared 10 wt.%. 

Lui and Sawant’s seminal work [3,51,52] characterized spherulitic supramolecular network structures of LMOGs using a fractal model combining the trimmed Caley Tree and Avrami models [53,54,55]. Self-similarity in fractal objects reflects the independence of length scale on its geometric properties and two-point density–density correlation function [3,56,57,58,59]. Three essential factors—the length between branch points, growth site activity, and each branching rate (z)—determine the type of Caley tree type and alter its geometric and physical properties (e.g., skeleton dimension) [51,60]. Networks characterized by Caley tree fractals maintain a constant branching rate (z), and newly formed ‘daughter’ segments maintain identical dimensions to the precursor ‘mother segment’, whereby perpetual recurrence of growth cycles forms the supramolecular hierarchical network [51,60,61]. DBS/PEG gels form trimmed Cayley Trees as they contain dead-ends, the branching rate, z, and segment length ζ, are constant throughout the crystal growth process, and branching occurs at all sites simultaneously [3,51,60]. At low magnifications (Figure 2 and Appendix A), only the supramolecular spherulitic network is apparent, and the fibers that comprise these networks become evident at much higher magnifications (Figure 3).

From images in Figure 3, surface PEG is displaced, exposing the DBS microstructure, illustrating the spherulitic crystallites seen as Maltese crosses (Figure 2) comprise fibers radially growing from the central nuclei, which is eventually impeded by the encroachment of adjacent crystals. These delineations between spherulites are most apparent in PEG200 after the 5 min water rinse and PEG400 and PEG600 after 15 min and are similar in dimension compared to the brightfield images of Figure 2. The individual, radially-growing fibers become exposed, and although visual differences are apparent, such as crystal size, the Cayley Tree fractal character is not obvious (Figure 3). At high magnifications, the fiber morphology in both 200PEG and 600PEG consists of ‘parent’ fibers, which branch into two ‘child’ fibers, ALL of which are equal in length and diameter (Figure 4).

At obvious branching points (Figure 4, white x), the branching rate is consistently two, and very little variation in fiber width across radial shells exists. Interestingly, the fibers which comprise each spherulite in PEG200 are approximately twice as wide for PEG600, at this point, it is not immediately obvious why such a drastic difference in fiber thickness presents. The solvent viscosity impacts the number of nucleation sights, spherulitic crystallite and fiber size, and the Caley tree fractal pattern (l and ζ) (Figure 4 and Appendix A). Remarkably, in addition to Cayley Tree fractal patterns presenting within the spherulite fibers, smaller fractal patterns appear on the fiber surface (Figure 4, far right). In an attempt to quantify the difficult-to-describe differences of the SAFiN morphologies across length scales, SEM images of the same DBS/PEG gels were obtained at 130×, 500×, and 2900× magnification (Figure 5). Post-acquisition modification included correcting the white balance to ensure the entire greyscale (0–255) was used before automated thresholding (percentile threshold function, Fraclac Plugin, ImageJ, NIH, Bethesda, MD, USA). In this case, Cayley Tree fractals present as a function of diffusion-limited growth, which, for obvious reasons, should be altered by solution viscosity. Diffusion-limited fractals often describe colloidal gels [62,63] and present fractal patterns that are statistically similar across length scales.

The box-counting fractal dimension, D_b_, is determined by plotting the count of foreground white pixels (N_ε_) in each different box size (ε) presented in Equation (1).
(1)$$D_{b}=\underset{\varepsilon \to \infty }{\lim}\left[\log N_{\varepsilon }/\log\varepsilon \right]$$

Albeit a small sample set (n = 3), D_b_ remains constant across magnifications for both 5 wt.% DBS in PEG200 and PEG600 (Figure 5), a requirement for fractal objects, but D_b_ is significantly lower for PEG600 than PEG200. Values of D_b_ ~1.8 indicate diffusion-limited cluster aggregation, while values ~2.8 suggest ballistic particle–cluster aggregation [64]. A key mechanism governing fiber–fiber interactions arises from crystallographic mismatches occurring at the crystal surface due to thermal fluctuation and mass transport properties [3,51,65,66]. SAFiNs formed at low super-saturations have large nucleation barriers. Thus randomness is suppressed at the growing surface, resulting in very high aspect ratio fibers [65]. High supersaturations suppress the mismatch nucleation barrier, leading to new crystalline domains on the crystal surface, resulting in deviations of the parent crystal orientation and fiber branching. When fiber branching occurs at the tip of the growing 1D fiber Cayley treelike, fractal networks are observed [3,67,68,69]. Cayley Trees are embedded in infinite Euclidian space; the total number of branching sites increases exponentially with R, the distance from the origin to the gth tree shell [51,61,70]. The exponent, D_f_ of Equation (2), represents the tree fractal dimension in the Euclidean space.
(2)$$M\;\sim {\;R}^{D_{f}}$$

The mass of the fractal (M) scales over chemical distance (ξ), the shortest vector between two branching sites and the chemical distance is not necessarily equal to the covering shells’ Euclidean distance represented by Equation (3) [60,61]: (3)$$M\;\sim {\;\xi }^{D_{\xi }}$$
where D_ξ_ is the fractal dimension in the chemical space and the fractal gyration radius (R) is obtained by the combination of Equations (2) and (3) [60,61]:(4)$$R\;\sim {\;\xi }^{D_{\xi }/D_{f}}$$

In Equation (4), the number of enclosed sites of the gth shell for a trimmed Caley tree pattern in the chemical space is represented as Equation (5) [51,60]:(5)$$M={\mathrm{zg}}^{D_{\xi }}$$
by considering the infinite distance of the gth shell in the fractal pattern (i.e., $\;g\;\sim \;{\left[\frac{R}{\xi }\right]}^{D_{\xi }/D_{f}},\;\mathrm{where}\;R\;\to \infty$), Equation (5) can be rewritten as Equation (2) for the fractal pattern. How solvent viscosity and density altered SAFiNs’ fractal dimension [51,61] was not elucidated until Liu and Sawant [52] modified the Avrami model (Equation (6)) [53,54,55] to report the fractal dimension (D_f_).
(6)$$\ln\left[1-X_{\mathrm{cr}}\right]=-{\mathrm{kt}}^{D}$$

The Avrami equation originally characterized the bulk crystal nucleation and growth rates, where k is a rate constant, t is time, and D indicates the bulk crystal growth dimension. Crystallinity (X_cr_) obtained from the crystal volume fraction, φ, at the time t, is divided by the φ as t → ∞ [51,53,54,55]: (7)$$X_{\mathrm{cr}}=\varphi \left[t\right]/\varphi \left[t\to \infty \right]$$

The Einstein relation, Equation (8), correlates X_cr_, or the volume fraction φ of suspended particles, to specific viscosity η_sp_ with the addition of a shape factor (F) [51], and η_sp_ is calculated from the complex viscosity of the system (η*) and initial solvent viscosity η_o_ [51,69,71,72,73,74], obtained from small deformation rheology (Figure 6A–C).
(8)$$\frac{n^{*}-n_{o}}{n_{o}}=\eta_{\mathrm{sp}}\approx F\varphi$$

Equation (9) combines Equations (6)–(8) and replaces D with D_f_, and t with (t − t_g_), where t_g_ is the sol-gel transition temperature, since the formation of a supersaturated state must precede nucleation and gel formation; and the ln{−ln [1 − X_cr_(t)]} versus ln(t − t_g_) (Figure 6D–F) corresponds to the fractal dimension D_f_ (Figure 6G,H).
(9)$$\ln\left[1-\left(\frac{\eta^{*}\left(t\right)-n_{o}}{\eta^{*}\left(\infty \right)-\eta_{o}}\right)\right]=-k{\left(t-t_{g}\right)}^{D_{f}}$$

The solvent viscosity has a significant impact on the fractal dimension (D_f_) across temperatures (Figure 6G), however, drastically different crystallization temperatures became less significant as solvent viscosity decreased (e.g., PEG200 presents the same D_f_ at all crystallization temperatures) (Figure 6H). This finding indicates that DBS SAFiN formation is governed by diffusion-limited aggregation (DLA) and not ballistic particle–cluster aggregation (BPCA). Additionally, DLA fractal values are ~1.8, compared to ~2.8 for BPCA [64], and, for the limited set of SEM images, the fractal values across magnifications remained mostly constant, ranging between 1.5–1.8, while the more robust rheological data set shows a similar decrease in fractal values between PEG200 and PEG600 where D_f_ ranged between 1.0–1.5 (Figure 6H). Absolute fractal values obtained from box-counting and the Avrami equation do not compare with different structural organization measures. However, the relative change in value for PEG200 and PEG600 shows a reduced fractal dimension of the DBS SAFiN in a higher viscosity environment.

## 3. Conclusions

Evidence of fractality in nature is always a remarkable phenomenon to report. Herein, DBS in PEG presents its fractality through the consistent D_b_, across magnifications, and through its Cayley Tree fractal branching pattern and corresponding linear Avrami determination of D_f_. Additionally, and somewhat surprisingly, the Cayley Tree patterns present for fiber branching within the spherulites and on the fiber branching on the individual fiber surfaces. Cayley Tree fractal fiber formation at two length scales (e.g., fibers in spherulites and fibers of fibers) has yet to be reported for SAFiNs. Solvent viscosity has a larger impact on supramolecular fractality than crystallization temperature, again supporting the notion that SAFiNs from diffusion-limited aggregation.

## 4. Materials and Methods

### 4.1. Sample Preparation 

1,3:2,4 dibenzylidene-D-sorbitol (DBS) (95%, BocSci, New York, NY, USA) and polyethylene glycol (PEG) 200 (mol. Wt. 190–210), 400 (mol. Wt. 380–420), and 600 (mol. Wt. 570–602), (Sigma Aldrich (Oakville, ON, Canada) were used as received. 5.0 wt.% samples of DBS in each polyethylene glycol (i.e., PEG 200, 400, and 600) were prepared in 2-mL borosilicate vials with Teflon-lined lids (VWR, Mississauga, ON, Canada), at 5 and 10 wt.%, and heated in an aluminum heating block until the sample was molten and transparent. Once molten, samples were held for 10 min and then held at 20 °C for 24 h (forced convection incubator, VWR, Mississauga, ON, Canada) or transferred directly onto the small deformation rheometer. 

### 4.2. Microscopy and Image Analysis

Brightfield and cross-polarized micrographs of the same field of view were obtained with an inverted microscope (Model CX41, Olympus, Tokyo, Japan) equipped with an imaging 5616 × 3744 pixels digital camera (Canon, Japan) and a 10× Olympus lens (0.25 N.A.) (Olympus, Tokyo, Japan). Samples placed on a glass slide were immediately covered with a coverslip and imaged. Crystal diameter was measured using ImageJ (NIH, Bethesda, MD, USA) and calibrated using a 10-μm scale bar. For scanning electron microscopy (SEM) 200 μL of 5 wt.% DBS/PEG molten gel was placed on the SEM stubs coated with carbon (600 Ultra Fine Norton SandWet^TM^, Worcester, MA, USA) and stored at room temperature for 24 h. Since PEG is more soluble in water than DBS, the stubs were soaked in ice-cold water and gently rotated. Three soaking times were applied to each gel (5, 15, and 30 min), after which they were dried at 40 °C for at least 2 h (Fisher Scientific, Isotemp^®^, Fair Lawn, NJ, USA) until the surface water evaporated. The stubs with gel were coated with a 20-nm thickness gold–palladium layer using a sputter coater (Denton vacuum Desk V, Moorestown, NJ, USA) operated with a 20-mA deposition current and vacuum pressure of 9 × 10^−5^ kPa. The gold–palladium-covered samples were placed in the SEM specimen holder (FEI Quanta FEG 250, Thermo Fisher Scientific, Hillsboro, OR, USA) and imaged with the xT Microscope Control software. An accelerating voltage of 5 kV was maintained. SEM images were prepared for thresholding by correcting the white balance in photoshop© (CS6, Adobe, San Jose, CA, USA) using the level function and setting the blackest and whitest points of the image to 0 and 255, normalizing image contrast. Images were then opened in ImageJ (1.52d, NIH, Bethesda, MD, USA) and thresholded using the percentile threshold function. The percentile threshold was manually adjusted, so SAFiN crystal features appear white, while all else appeared black. Images were then analyzed using the box-counting (D_b_) function in FracLac (V2.0f, Wagga Wagga, Australia) plugin for ImageJ. The D_b_ fractal dimension was calculated using 100 grid sizes between 20 pixels and 35% of the binary image, and the background was locked to black. Comparisons of statistical significance for all parameters compiled a minimum of triplicates using two-way ANOVA and Tukey’s Post hoc analysis (GraphPad Prism V.9) at *p* ≤ 0.05.

### 4.3. Small Deformation Rheology

Small-deformation rheology on a Physica MCR 301 rheometer equipped with a temperature-controlled Peltier plate and stainless-steel, 50-mm flat-parallel plate attachment (PP50/P2) (Anton-Paar, Graz, Austria) obtained parameters, storage (G′) and loss (G″), and complex (G*) moduli at 0.1% strain (γ) and 10 s^−1^ frequency controlled strain applied during temperature sweeps from 110 °C to either 20, 30, 40, 50, or 60 °C at 20 °C/min and then held isothermally until the G′ and G″ plateaued. 

## Figures and Tables

**Figure 1 gels-07-00046-f001:**
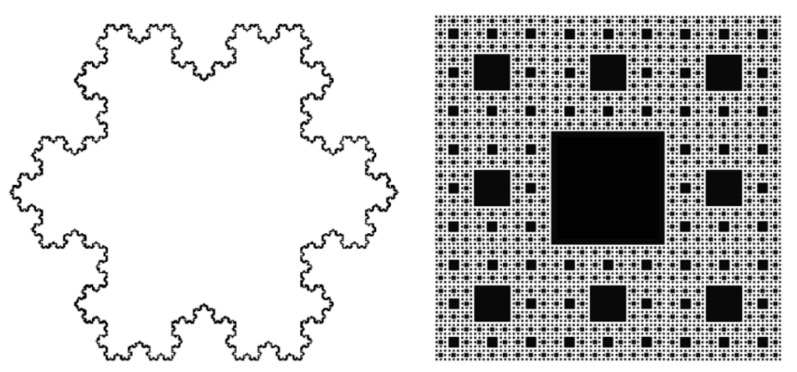
Illustration of self-similar fractal objects—the Koch Snowflake and the Sierpinski carpet.

**Figure 2 gels-07-00046-f002:**
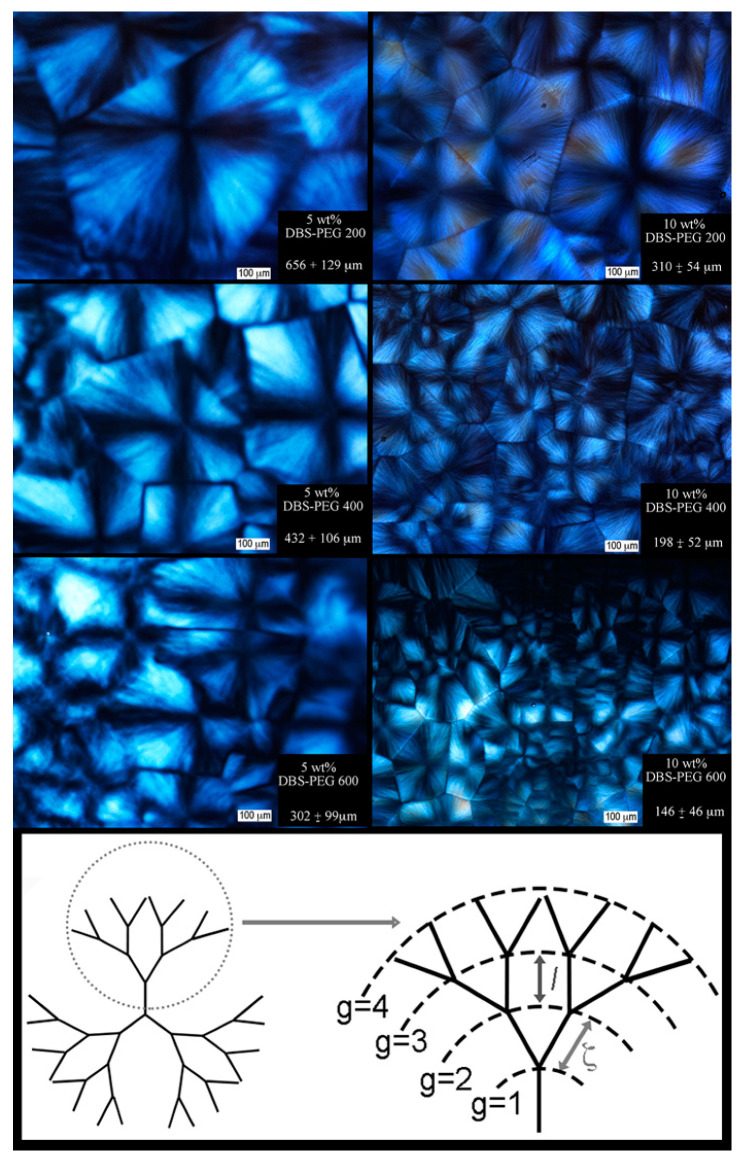
Polarized light micrographs of 2,4-Dibenzylidene sorbitol (DBS) in polyethylene glycol (PEG) 200, 400, and 600 crystallized at 20 °C. Crystal size average ± standard deviation (n = 3) (10 crystals per micrograph and if <10 crystals present, all were measured, each crystal was measured across the longest and shortest widths). Illustration of Trimmed Caley Tree.

**Figure 3 gels-07-00046-f003:**
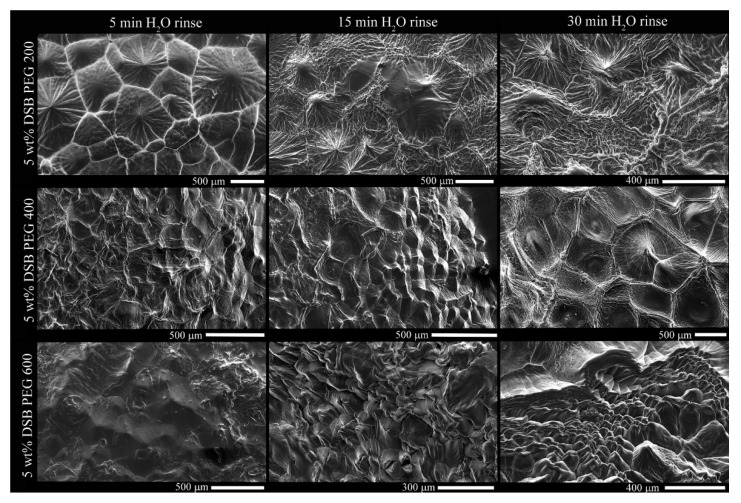
Scanning electron microscopy (SEM) micrographs after 5, 15, and 30 min (**Left**, **Center**, and **Right** columns) H_2_O rise to displace surface PEG (PEG200 (**Top row**), PEG400 (**Middle row**), and PEG400 (**Bottom row**)).

**Figure 4 gels-07-00046-f004:**
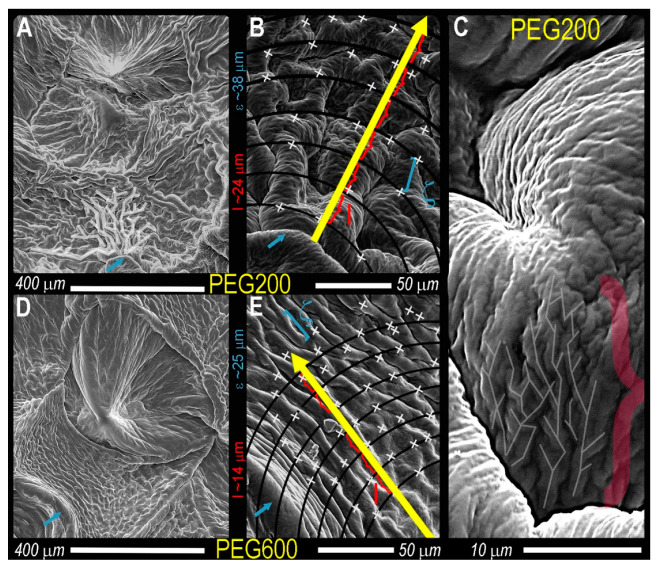
SEM images of trimmed Caley tree fractals present in 5 wt.% DBS in PEG200 (**A**–**C**) and PEG600 (**D**,**E**). Blue arrows represent the same point on high and low magnification images. The yellow arrow is the direction of the radial growth, where x represents branch points, the length (red), and chemical distances. Lightened lines in A and C are to illustrate the branched network.

**Figure 5 gels-07-00046-f005:**
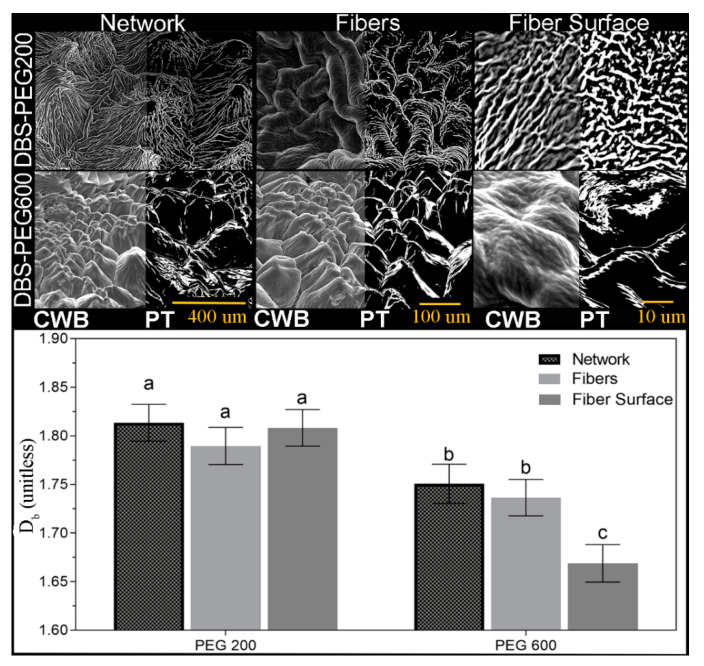
Corrected White Balance (CWB) and Percentile Thresholded (PT) of 5 wt.% DBS in PEG 200 or PEG600 SEM images across magnifications and the images resulting in box-counting fractal dimensions (D_b_). Letters above bars represent significant differences (*p* < 0.05).

**Figure 6 gels-07-00046-f006:**
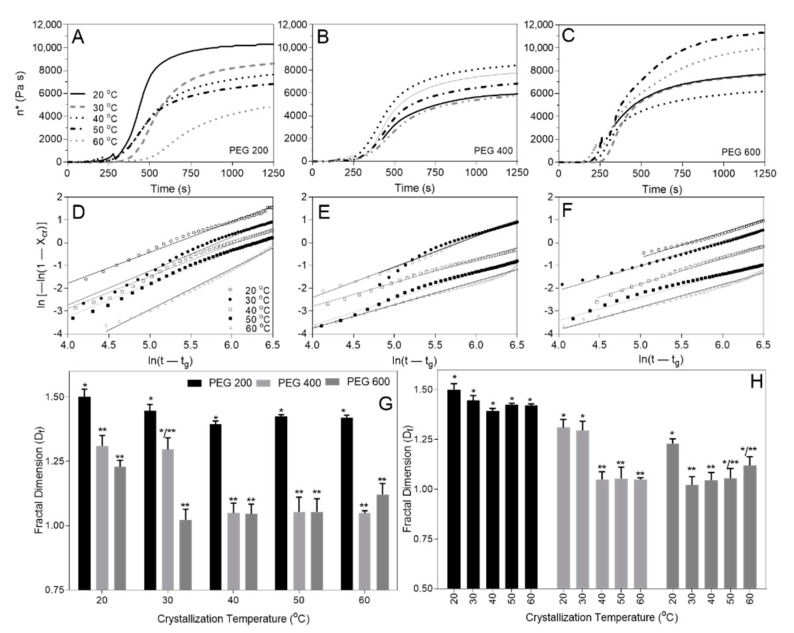
Complex viscosity (**A**–**C**) and correspondant ln{−ln[1 − X_cr_(t)]} versus ln(t − t_g_) (**D**–**F**) for PEG200 (**A**,**D**), PEG400 (**B**,**E**), and PEG600 (**C**,**F**). Fractal dimensions clustered by crystallization temperature (**G**) and PEG solvent (**H**). Different number of superscript stars * represents statical significance at *p* ≤ 0.05.

## Data Availability

Data is contained within the article and Appendix A. Additional images are available in the Appendix A. The data will be gladly shared in Prism or Excel Formats on request.

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
