# Peer review of "Supramolecular Fractal Growth of Self-Assembled Fibrillar Networks"

_gels, 2021, doi:10.3390/gels7020046_

Round 1

Reviewer 1 Report

Manuscript “Supramolecular Fractal Growth of Self-Assembled Fibrillar Networks” by Pedram Nasr , Hanna Leung, France-Isabelle Auzanneau, and Michael A. Rogers submitted to Gels represents the investigation of self-assembly of DBS molecules in PEG solvent into fibrillary network. Authors studied the effect of solvent viscosity on structure of formed network and rheological properties. It looks that solvent viscosity (molecular weight of PEG) is more important than Temperature. The structure of formed fibrils can be described by fractal objects which has been investigated by imaging techniques (polarised microscopy) and SEM. Rheology is done by small deformation approach. The subject of research is important as similar self-assembly is observed in different parts of modern science from chemistry to biology. On my mind, authors should carefully checked obtained results and try to connect obtained 2D images with 3D viscosity of solution. May be add some scattering method to get 3D structure (SAXS?). Authors have got G´ and G´´ but it was even not mention in discussion. Some other points, which should be corrected:

  1. In 1st paragraph Df for Koch Snowflake is wrong. Please check and correct.
  2. Scale bars in Figure 3 are missing and in Figure 4 are bad visible.
  3. In Eqs. Symbol sometimes are not determined or it happens in few Eq. later.
  4. Please provide list of abbreviation and symbols.
  5. In row 99 Authors claim that Cayley tree fractal character is not obvious and authors did not explain why it is used in further analysis.
  6. In row 54, author mention poly(propylene glycol) and it is missing in data section and in other part of manuscript.   

Author Response

Manuscript “Supramolecular Fractal Growth of Self-Assembled Fibrillar Networks” by Pedram Nasr , Hanna Leung, France-Isabelle Auzanneau, and Michael A. Rogers submitted to Gels represents the investigation of self-assembly of DBS molecules in PEG solvent into fibrillary network. Authors studied the effect of solvent viscosity on structure of formed network and rheological properties. It looks that solvent viscosity (molecular weight of PEG) is more important than Temperature. The structure of formed fibrils can be described by fractal objects which has been investigated by imaging techniques (polarised microscopy) and SEM. Rheology is done by small deformation approach. The subject of research is important as similar self-assembly is observed in different parts of modern science from chemistry to biology.

On my mind, authors should carefully checked obtained results and try to connect obtained 2D images with 3D viscosity of the solution.

This is a contentious point across colloidal fractals, and to be frank, the images and rough fractal calculations are done more in a supportive role to the more robust rheological data set showing rough branching patterns supporting Liu and Sawant's seminal work.  There are so many confounding factors when trying to compared 2 vs 3 D fractals- well beyond the scope of the author's expertise.

Authors have got G´ and G´´ but it was even not mention in discussion. Some other points, which should be corrected:

 The G’ and G’’ are used to calculate the complex viscosity, a G’ and G’ vst Time profile and N* vs time it is repetitive.

  1. In 1st paragraph Df for Koch Snowflake is wrong. Please check and correct.

It has been corrected to Df in line

  1. Scale bars in Figure 3 are missing and in Figure 4 are bad visible.

Thank you, we have improved the qualaity of the images, they did not scale well when inseted into the manuscript.

  1. In Eqs. Symbol sometimes are not determined or it happens in few Eq. later.

They are all corrected in lines 123, 145, 149, 152, 155, 161, 166, 171 and 176.

  1. Please provide list of abbreviation and symbols.

The list is provided and is added to the supplementary part.

List of Symbols and Abbreviations

SAFiNs

self-assembled fibrillar networks

DBS

1,3: 2,4-Dibenzylidene sorbitol

PEG

polyethylene glycol

Df

fractal dimension

Db

box-counting fractal dimension

d

LMOGs

low-molecular-mass organogelators

PPG

polypropylene glycol

z

branching rate

z

Segment length

l

distance within the Euclidian embedding space

Ne

the count of foreground, white pixels

e

box size

PT

Percentile Threasholded

CWB

Corrected White Balance

BPCA

ballistic particle–cluster aggregation

g

number of fractal shell in Caylee tree

R

the distance from the origin to the gth tree shell

M

Mass of the fractal

ξ

Chemical distance

Dξ

Fractal dimension in chemical space

Xcr

Crystallinity

t

Time

D

bulk crystal growth dimension

k

Rate constant

j

crystal volume fraction

hsp

specific viscosity

F

Shape factor

h*

complex viscosity of the system

ho

initial solvent viscosity

  1. In row 99 Authors claim that Cayley tree fractal character is not obvious and authors did not explain why it is used in further analysis.

The fractal character is not even obvious in that resolution (magnification) and yet needs to go to higher ones as in Figure 4. However, the authors tried to address the misunderstanding of the sentence in line 97.

  1. In row 54, author mention poly(propylene glycol) and it is missing in data section and in other part of manuscript.  

Polypropylene glycol (PPG) spelling is fixed in lines 54 and 55. In this manuscript we did not used PPG and thus, there is no new data obtained. However, we did a comparison between gels in PEG and published literature for the gels in PPG.

Reviewer 2 Report

Review of gels 1138667

This manuscprit describes the dissolution and recrystalization of 2,4,-dibenzylidensorbitol (DBS) in polyethylenglycols (PEGs) of three different (low) molecular weights. Upon cooling and crystallization, DBS in PEG forms certain structures / crystallites, in which the authors observe fractal dimensions. The geometry of these fractals and spherulites is analyzed through the processing of light microscopy and SEM images. Correlations are made with the fractal dimensions and the rheological properties of the PEG/DBS solutions. The main finding is, that lower Mw PEG, which has a lower viscosity, causes the growth of larger spherulites, which also show slightly higher fractal dimensions (self similarity when increasing resolution) than those observed in higher Mw PEG. A diffusion limited process of structure formation is suggested.

I think the work was designed with care and the interpreation of data and the findings have some significance for the understandig of PEG gels and crystallization of polyol derivatives.

It can therefore be accepted after the consideration of some comments.

  1. How representative are the images and the sample size? Line 244 states triplicates were used. Are these triplicates of images of one composition or triplicate preparations? When looking at the images in Fig. 3 / 4 / 5, I could assume that replications could deviate easily. Finally it is crystallization, and known that repetitions can be quite difficult. Sample numbers should be better specified in the experimental, and if possible several images of replicates shown in supplementary data.
  2. Figure 2: If the branching tree forms like this, would it be better seen when the centre of the crystallization is controlled and varied, e.g. by dropping solutions at certain points of a surface and check if they converge? The roots and branches of the tree could be marked in the images so as to better understand how the authors interpret them. Same in Fig 3.
  3. Figure 5: should read: unitless.
  4. Line 147: how large do the authors estimate the chemical distances? The length of the PEG or beyond?
  5. Figure 6: Would benefit from comparing the complex viscosities of pure PEGs (maybe in supplementary) and 10% DBS in PEGs and some interpretation of the molecular interaction of DBS/PEGs.
  6. Line 211-213: was 5wt% DBS used or 5wt% and 10 wt% ? (line 2013).
  7. Line 215: how would the holding time incl. the agitation during that time influence the structure of the gels?

Author Response

Review of gels 1138667

This manuscprit describes the dissolution and recrystalization of 2,4,-dibenzylidensorbitol (DBS) in polyethylenglycols (PEGs) of three different (low) molecular weights. Upon cooling and crystallization, DBS in PEG forms certain structures / crystallites, in which the authors observe fractal dimensions. The geometry of these fractals and spherulites is analyzed through the processing of light microscopy and SEM images. Correlations are made with the fractal dimensions and the rheological properties of the PEG/DBS solutions. The main finding is, that lower Mw PEG, which has a lower viscosity, causes the growth of larger spherulites, which also show slightly higher fractal dimensions (self similarity when increasing resolution) than those observed in higher Mw PEG. A diffusion limited process of structure formation is suggested.

I think the work was designed with care and the interpretation of data and the findings have some significance for the understandig of PEG gels and crystallization of polyol derivatives.

It can therefore be accepted after the consideration of some comments.

  1. How representative are the images and the sample size? Line 244 states triplicates were used. Are these triplicates of images of one composition or triplicate preparations? When looking at the images in Fig. 3 / 4 / 5, I could assume that replications could deviate easily. Finally it is crystallization, and known that repetitions can be quite difficult. Sample numbers should be better specified in the experimental, and if possible several images of replicates shown in supplementary data.
    We agree and the process being stochastic being a confounding factor.  The differences in networks are added with composite images in SI.
  2. Figure 2: If the branching tree forms like this, would it be better seen when the centre of the crystallization is controlled and varied, e.g. by dropping solutions at certain points of a surface and check if they converge? The roots and branches of the tree could be marked in the images so as to better understand how the authors interpret them. Same in Fig 3.  Agreed, thank you,  I had tied to illustrate it in figure 4 and have redone to try and improve the presentation.
  3. Figure 5: should read: unitless. Thank you for catching this typo, it has been corrected.
  4. Line 147: how large do the authors estimate the chemical distances? The length of the PEG or beyond?

The chemical distance in the Cayley tree is the shortest distance between brach points, in all cases this is in the microscale and thus far beyond the length of DBS.  Each fiber chemical distance contains 100-1000s of DBS molecules.

  1. Figure 6: Would benefit from comparing the complex viscosities of pure PEGs (maybe in supplementary) and 10% DBS in PEGs and some interpretation of the molecular interaction of DBS/PEGs.

This was the hold up for the revision.  We agreed with the comment and set up experiments to complete;  however, since complex viscosity is obtained from G’ and G’’ the liquid measurements have spindle inertia artifacts and is why the fractal dimension slopes are started at the melting temperature of the gel.

  1. Line 211-213: was 5wt% DBS used or 5wt% and 10 wt% ? (line 2013).

5 and 10 wt% were used and it is corrected in lines 210 and 212.

  1. Line 215: how would the holding time incl. the agitation during that time influence the structure of the gels?

It is a good question, the agitation in small deformation rheology is trivial and not sufficent to sigificantly alter cyrstallization, especially aggregate formation. A significant shear force would be required.  Additionally, our parameters are selected based on the Linear viscoelastic region and are not expected to alter the overall elasticity of the network.

Round 2

Reviewer 1 Report

Authors have tried to improve manuscript but there are still some points which should be done better.  The next points should be done:

1. Authors have understood wrongly my previous remark (1),

I mean that "In 1st paragraph Df for Koch Snowflake is wrong." The numerical expression is wrong.  It is written in manuscript, "(Df =1.26 (D = 4ln(3))" but it is not correct. It should be log(4)/log(3). 

2. Authors should check Fig. 3, the text on Figure is written in wrong direction.

3. Authors should inlclude list of abbriviation into manuscript. In other case it is impossible to read.

Author Response

  1. Authors have understood wrongly my previous remark (1),

I mean that "In 1st paragraph Df for Koch Snowflake is wrong." The numerical expression is wrong.  It is written in manuscript, "(Df =1.26 (D = 4ln(3))" but it is not correct. It should be log(4)/log(3).

We apologize and we have corrected the formula, thank you.

  1. Authors should check Fig. 3, the text on Figure is written in wrong direction.

We transformed the landscape image to a portrait so the text is more suitable.

  1. Authors should include list of abbreviation into manuscript. In other case it is impossible to read.

Sorry, we had added it to the supplemental information but it is now located in the manuscript.